# A structured medication review tool to promote psychotropic medication optimisation for adults with intellectual disability: feasibility study

Rory Sheehan [iD],[1] André Strydom,[2] Louise Marston,[3] Nicola Morant [iD],[1] Federico Fiori,[4] Paramala Santosh [iD],[5] Angela Hassiotis[1]

[1]Division of Psychiatry, University College London, London, UK
[2]Department of Forensic and Neurodevelopmental Science, Institute of Psychiatry, Psychology & Neuroscience, London, UK
[3]Research Department of Primary Care and Population Health, University College London, London, UK
[4]Department of Child and Adolescent Psychiatry, Institute of Psychiatry, Psychology & Neuroscience, London, UK
[5]Department of Child and Adolescent Psychiatry, King's College London, London, UK

**Correspondence to**
Dr Rory Sheehan;
r.sheehan@ucl.ac.uk

## ABSTRACT

**Objectives** To investigate the feasibility of delivering structured psychotropic medication review in community services for adults with intellectual disability (ID).

**Design** Single-arm feasibility study conducted over a 6-month period.

**Setting** Specialist community ID teams in England.

**Participants** Psychiatrists working with adults with ID and adults with ID who had been prescribed psychotropic medication.

**Intervention** A structured web-based psychotropic medication review tool (the HealthTracker-based structured medication review) comprising measures of therapeutic benefit and adverse side-effects was made available for use by psychiatrists in routine clinic appointments. A summary measure of medication effectiveness was graphically presented to aid discussion and decision-making.

**Main outcome measures** Feasibility metrics including number of people with ID referred, eligible and recruited, and uptake of the medication review tool in naturalistic clinical settings. Psychiatrist and patient feedback was collected to assess acceptability of the intervention and suggestions for development.

**Results** Fifteen psychiatrists from five clinical teams took part. In total 94 potentially eligible people with ID were referred, of whom 79 (84%) were recruited and together underwent 97 medication reviews over the 6-month study period. Feedback from participants with ID was favourable. Psychiatrists indicated that the HealthTracker-based structured medication review was broadly acceptable and suggested adaptations to improve integration with existing information technology systems and to enhance patient involvement in the review.

**Conclusions** Structured psychotropic medication review can be used in community services for adults with ID as part of a programme of medication optimisation. It would be feasible to test clinical and patient outcomes of the HealthTracker-based medication review in a randomised clinical trial.

## INTRODUCTION

Intellectual disability (ID), present in approximately 2% of the population, is a lifelong disorder defined by significant cognitive

### Strengths and limitations of this study

► This feasibility study is one of the earliest to suggest a pragmatic and scalable means of achieving psychotropic medication optimisation in people with intellectual disability using structured medication review.
► The work provides estimates of recruitment rate and uptake of the intervention, as well as suggestions for its development, that can inform the planning and delivery of a future clinical trial.
► The study was conducted in a single region of the UK which may not be representative of other locations or healthcare settings.
► Details of those who were eligible but did not participate in the study were not collected and the overall rate of uptake of the intervention cannot be determined.

deficit and impaired functional and adaptive skills.[1] Between one-third and one-half of adults with ID are prescribed psychotropic medication.[2][3] Renewed focus on the quality of prescribing has been prompted by epidemiological evidence which shows that the extent of psychotropic use is disproportionate to prevalence of mental illness in this group, and medication is often used 'off-label' in the management of behaviour that challenges.[4] People with ID are at greater risk of idiosyncratic reactions and adverse medication side-effects than their non-intellectually disabled counterparts and are more likely to receive high psychotropic doses, polypharmacy, and to remain on psychotropic medication for extended periods.[5][6]

The UK Government has committed to improving the use of psychotropic medication in people with ID[7] and a national programme, Stopping the Over-Medication of People with Learning Disabilities (STOMP), was established in 2016 to raise awareness

of the issue and stimulate activity among patients, advocates and professionals.[8] Medication optimisation is a multifaceted concept that aims to promote the best use of prescribed medication by prioritising safety, evidence-based choice of medication and centring patient experience and involvement.[9] Medication review, a structured and critical evaluation of a prescribed medication, is a key element of medication optimisation that is recommended by the National Institute for Health and Care Excellence for groups at high risk of suboptimal medication use.[10] Structured medication review offers a number of potential benefits including promoting systematic evaluation of desired and undesired medication effects; standardisation of assessment across time and between clinicians; an efficient method of recording information and making explicit the basis on which decisions are made. A recent systematic review found that psychotropic medication review is associated with change or reduction in number of drugs prescribed but consistent improvement in clinical and patient-reported outcomes has not been shown, and there is considerable variation and little formal guidance on how medication reviews are operationalised.[11] We undertook a study to investigate the feasibility of a structured psychotropic medication review (the HealthTracker-based structured medication review (HT-SMR)) in community psychiatry of ID teams. Specific objectives were to determine the recruitment rate of psychiatrists and people with ID to the study, to assess the uptake of this novel intervention in real-world clinical settings and to gather feedback that could inform future development of the intervention.

## METHOD
### Study procedures
This was a single-arm feasibility study conducted over a 6-month period in five community psychiatry of ID services in London, UK. All services were part of the National Health Service. The study and its rationale were presented to psychiatrists in participating clinical teams, and they were then invited to take part in the study. If they agreed, they were given access to the HT-SMR for the study period. Adults (>18 years) with ID were eligible to participate if they were prescribed psychotropic medication of any type and for any indication. Psychiatrists were asked to briefly introduce the research to potential participants and/or their carers, either at routine appointments or by sending an information leaflet through the post. The contact details of those who expressed interest were passed to the research team who then met with the potential participant to explain the research in more detail and confirm eligibility. Written informed consent was obtained from all people with ID. Ability to consent to take part was assessed according to the principles of the Mental Capacity Act.[12] If a person lacked capacity to consent, a family member or nominated consultee was sought to give advice to the research team on the person's inclusion. All study materials were available in accessible

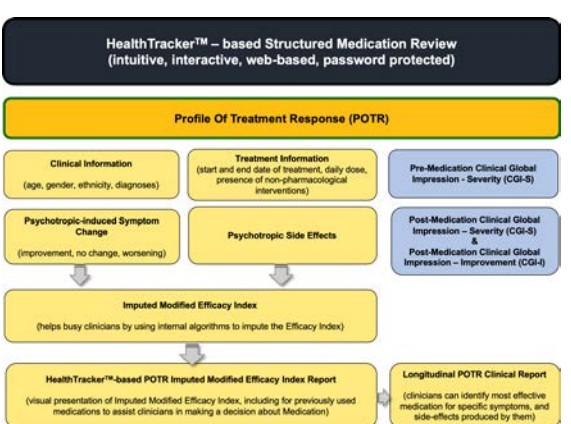

**Figure 1** HealthTracker-based structured medication review.

(easy-read) format. When a participant was recruited to the study, his or her psychiatrist was informed and was then able to use the HT-SMR in appointments with that person.

### Intervention
The intervention consisted of the HT-SMR (figure 1) designed to be used in routine clinical appointments by a participant's psychiatrist. The HealthTracker is a password-protected web-based health-monitoring platform that originated in the NHS, with the NHS receiving royalties from its use. For the purposes of this study, medication review included a record of basic demographic, clinical and treatment information along with responses to the Profile of Treatment Response (POTR). The POTR comprises two generic scales: one measuring therapeutic response to a medication over several symptom domains and the other measuring potential adverse side-effects. Each item is rated by the psychiatrist on a Likert scale using information gathered from observation and the clinical interview. Items that are not applicable can be marked as such but incomplete reviews cannot be submitted. Based on responses to the two scales above, the HealthTracker imputes the Modified Efficacy Index (MEI) as the ratio between the therapeutic benefit of a medication and the presence of adverse side-effects. The MEI is then displayed in a simple colour-coded matrix that allows viewers to see how the patient has responded to treatment and may act as a stimulus for discussion between the psychiatrist and the patient and/or carer. The Clinical Global Impression-Improvement (CGI-I), a well-established rating tool that can be completed quickly and easily in clinical settings,[13] is completed by the psychiatrist for each medication as a further measure of medication effect. We asked psychiatrist to record if they had advised a change to medication following the review.

Each participant with ID was assigned a unique identification number and pseudonymised data collected in the medication review were stored on a secure electronic cloud. A single medication or multiple medications could be reviewed at one time, with a separate POTR and separate CGI-I for each drug that was reviewed. If the HT-SMR

is used across different time periods, a longitudinal record of treatment response to a certain medication is generated. The researcher trained psychiatrists on using the system in face-to-face small-group sessions focused on the practicalities of opening a case and entering data, and used a fictional patient to reinforce the learning. The research team were available for support as needed throughout the study.

Data from medication reviews were downloaded from the HealthTracker as a CSV file into SPSS V.24 at the end of the study period. The POTR, MEI and CGI-I results were summarised with descriptive statistics. Spearman's correlation between the MEI and CGI-I and the psychiatrist's decision to change or not to change medication was calculated. Owing to the skewness of the data, non-parametric tests were used to test the significance of associations.

### Feasibility measures

We gauged interest from clinical teams and individual psychiatrists to take part in the study and recorded the rates of referral and recruitment of people with ID, and of uptake of the medication review tool in routine clinic appointments. Reasons for not recruiting people who were referred to the research team were noted. As this was a feasibility study, a formal sample size calculation was not performed but our a priori estimate was that 100 people with ID would be recruited based on previous feasibility studies that have trialled similar interventions in community settings.[14]

### Participant characteristics

Characteristics of people with ID who were recruited and descriptive data concerning diagnosis and medication use are reported. Medication doses were converted to defined daily dose (DDD).[15]

### Acceptability and implementation

At the end of each medication review, psychiatrists asked people with ID, "How able were you to say everything you wanted to say about medication today?" Answers were scored on a 5-point Likert scale with pictorial cues alongside the response set to improve understanding. At the end of the study period, psychiatrists were invited to complete an anonymous web-based survey designed for this study with a mix of closed and open-ended questions. The survey concerned the research process, experience and views on use of the online review system, and suggested adaptations to maximise usability and utility of the medication review in its future development. Responses to the psychiatrist feedback questionnaire were summarised in a structured analysis within pre-determined categories. All data were managed in SPSS V.24 and Microsoft Excel.

### Patient and public involvement

A service user consultation group was formed as part of the wider programme of work within which this study was conducted. The consultation group consisted of people with ID and experience of medication use. We held regular meetings with the consultation group, who advised on various aspects of this work including the recruitment strategy, participant materials and easy-read information, devising the outcome measure for participants with ID, and general advice to aid successful conduct of the research. The group will be involved in dissemination to a broad range of relevant stakeholders.

## RESULTS

### Recruitment and uptake

Five community ID teams comprising 15 psychiatrists were invited and agreed to take part in the feasibility study which was conducted between September 2018 and March 2019. Eight psychiatrists were of consultant grade (who had completed specialist training in psychiatry of ID) and seven psychiatrists were trainees (with between 6 months and 3 years experience working in with people with ID). Together, 94 people with ID were referred as potential participants over the 6-month study period and 79 (84%) were recruited. Psychiatrists used the online system for medication review in 68 people (86% of those recruited). A number of people (n=21) had more than one medication review (when either more than one medication was reviewed at a single time point, or a single medication was reviewed on more than one occasion) giving a total of 97 HT-SMRs (figure 2).

There was a steady state of referral, recruitment and review tool use (figure 3A). Recruitment and uptake of the HT-SMR was unequal between participating community ID teams and not related to the number of psychiatrists in each of these teams (figure 3B). Each psychiatrist conducted a median of 7 medication reviews using the HT-SMR (range 0–20). No harms or unintended consequences were reported during the study and no participants withdrew their consent.

### Participant information and data from medication reviews

Demographic data of participants with ID who had medication review are summarised in table 1. The group was relatively young and most had mild ID. A primary diagnosis was not recorded in just over half of the participants; in these cases, it is possible that psychotropic medication was prescribed for behaviour that challenges.

Of the 97 HT-SMRs conducted using the system, the most commonly reviewed drug class was antipsychotics (49 reviews), followed by antidepressants (28 reviews) (table 2). The median prescribed dose of medication reviewed was 100% DDD (interquartile range (IQR) 50%–133%) and median duration of use was 18 months (IQR 5–56 months). Following the HT-SMR, psychiatrists advised a change to medication in just over one-third (n=27, 36%) cases.

The HealthTracker-imputed MEI can take a value of 0.33–4.0, where higher values equate to a more favourable therapeutic effect:adverse side-effect ratio. The median HealthTracker-imputed MEI for medications reviewed was 1.5 (IQR 1.0–3.0). There was a statistically

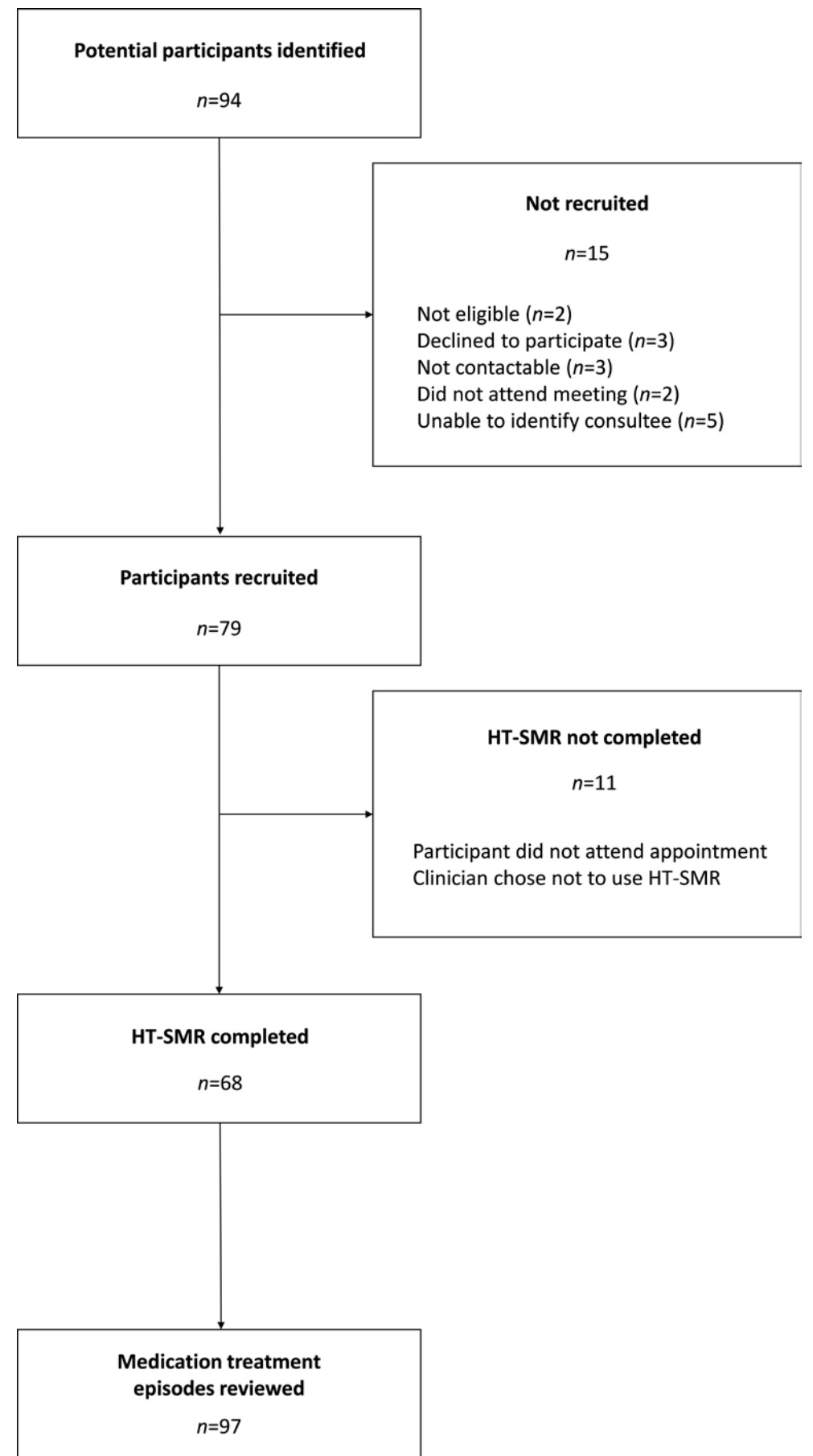

**Figure 2** Participant flow. HT-SMR, HealthTracker-based structured medication review.

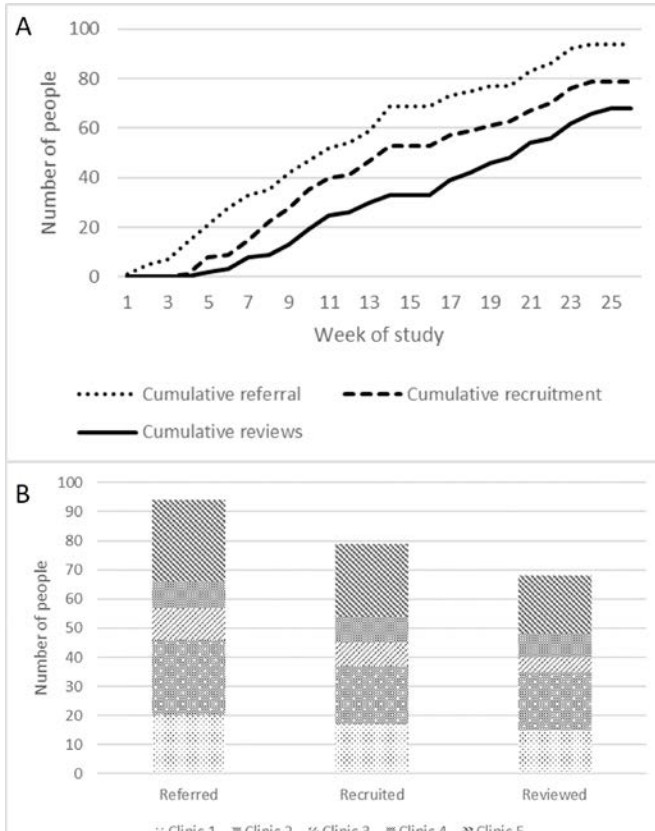

**Figure 3** (A) Rate of referral, recruitment and use of the HT-SMR over the study period and (B) by participating clinical team.

| Table 1 Demographic characteristics of participants with ID | |
|---|---|
| **Characteristic** | **n (%)** |
| Sex | |
| Male | 41 (60) |
| Female | 27 (40) |
| Age at first HT-SMR (years) | |
| 18–25 | 23 (34) |
| 26–35 | 16 (24) |
| 36–45 | 8 (12) |
| 46–55 | 17 (25) |
| 55–65 | 1 (1) |
| >65 | 3 (4) |
| Degree of ID | |
| Mild | 42 (62) |
| Moderate | 18 (26) |
| Severe-profound | 8 (12) |
| Ethnicity | |
| White | 35 (51) |
| Black | 14 (21) |
| Asian | 10 (15) |
| Mixed/other | 7 (10) |
| Not known/not given | 2 (3) |
| Primary diagnosis | |
| Schizophrenia spectrum disorder | 12 (18) |
| Mood disorder | 5 (7) |
| Anxiety disorder | 3 (4) |
| Personality disorder | 1 (1) |
| Pervasive developmental disorder | 7 (10) |
| Attention deficit hyperactivity disorder | 3 (4) |
| Missing | 37 (54) |

HT-SMR, HealthTracker-based structured medication review; ID, intellectual disability.

significant negative correlation between the MEI and the CGI-I (where a lower score indicates greater perceived benefit of medication) (r –0.296, p=0.024; indicating 'fair' correlation between the two measures).[16] The MEI was significantly lower in those in whom a medication change made following the review (median MEI 1.0, IQR 0.67–2.0) compared with those in whom no medication change was made following the review (median MEI 1.5, IQR 1.3–3.0) (p=0.011).

### Acceptability and implementation

When asked "How able were you to say everything you wanted to say about medication today?", participants with ID responded 'very easy' or 'easy' in 54 (70%) cases, 'not easy or difficult' in 14 (18%) cases, 'difficult' or 'very difficult' in 1 (1%) case. The question was not answered by 9 (12%) participants with ID.

Majority of psychiatrists (14/15) completed the online feedback questionnaire. Results are presented as major themes with anonymised quotations to illustrate points of interest.

### Feedback about the recruitment process

Although the majority (13/14) of psychiatrists reported that it had been 'easy' to introduce the study to potential participants, most of them (11/14) had encountered barriers. The main barriers to recruitment were 'time constraints' within appointments and difficulties

explaining the research to potential participants, especially those with more severe ID.

Psychiatrists were asked if the people who did not wish to hear more about the research had given reasons for their decision. The most commonly reported reason (eight cases) was worry about the commitment or inconvenience the research would entail. Others declined to hear more as they were already taking part in research or were content with their current medication regimen and did not want to discuss this further. Seven psychiatrists reported that the person's carer had not wished to pursue the research opportunity, either because they felt it was not appropriate or because they were not willing to act as a consultee in cases where the person with ID was likely to lack capacity to provide informed consent.

**Table 2** Summary results from the HT-SMRs (n=97 reviews)

| Drug class reviewed | Number of reviews (% of all reviews) | Median DDD of medication reviewed (IQR) | Median duration of use (months) (IQR) | Median CGI-I (IQR)* | Median Modified Efficacy Index (IQR)† |
|---|---|---|---|---|---|
| Antipsychotic | 49 (51%) | 67 (45–100) | 24 (4–60) | 1.5 (1.0–2.0) | 2.0 (1.0–3.0) |
| Antidepressant | 28 (29%) | 150 (87–200) | 12 (4–24) | 2.0 (1.0–3.0) | 2.0 (1.4–3.0) |
| Anxiolytic/sedative | 9 (9%) | 24 (2–54) | 100 (39–100) | 3.0 (2.0–3.0) | 3.0 (2.3–4.0) |
| Medication for attention deficit hyperactivity disorder | 9 (9%) | 120 (78–138) | 18 (12–78) | 2.0 (1.0–2.5) | 2.0 (1.0–4.0) |
| Mood stabiliser | 2 (2%) | 33 | 96 | 2.5 | 1.2 |
| All | 97 (100%) | 100 (50–133) | 18 (5–56) | 2.0 (1.0–3.0) | 1.5 (1.0–3.0) |

*CGI-I is scored between 1 (very much improved) and 7 (very much worse).
†Modified Efficacy Index is the ratio between the score in the domain with the greatest therapeutic benefit and the score in the domain with the worst rated adverse side-effect. Higher scores indicate more favourable medication response.
ADHD, Attention Deficit Hyperactivity Disorder; CGI-I, Clinical Global Impression-Improvement; DDD, defined daily dose; HT-SMR, HealthTracker-based structured medication review; IQR, interquartile range.

### Ease of using the HealthTracker online system

Twelve psychiatrists reported having used the online system for medication review. In response to the question, "How easy was it to use the HealthTracker?" only one person reported it was 'difficult'; the majority said it was 'easy' or 'very easy'.

### Benefits of HT-SMR

Eight out of 12 psychiatrists were of the opinion that using the online medication review had helped people with ID or their carer to be more involved in the discussion about medication and promoted 'collaborative decision-making'. Psychiatrists commented that it had been 'helpful as a template' in 'framing the discussion around medication' and that its use facilitated 'more in-depth' and 'comprehensive' medication review. The transparency of medication review using the system was described as an advantage: '[using the tool] was an eye-opener for the patient and carer. They could hear the specific questions being asked systematically and seeing the matrix [the graphical representation of the EI] was really useful, particularly for a few participants who were not clear about medication'.

### Disadvantages of the HT-SMR

Six psychiatrists described logistical problems in using a system which required internet connectivity, for example, computers not working or running fast enough, lack of internet access in clinic settings, not having portable devices for domiciliary visits. Using the system took additional time which was sometimes difficult to find in the regular appointment.

Just over half (7/12) of psychiatrists expressed the view that using the online medication review had interfered with their interaction with the patient or carer with one remarking that they had spent 'more time focussed on the computer rather than face-to-face personal interaction'. Two psychiatrists considered the system too rigid and resisted the 'imposed structure' of the medication review which they believed was not always aligned with the patient's most pressing concerns.

### Effect on decision-making

Eight out of 12 psychiatrists thought that undertaking the HT-SMR had helped them to make a decision about medication and 5/12 considered the tool made it more likely they would change medication compared with their usual practice. However in the survey free-text responses most commented that the medication review did not cause them to change decisions they would ordinarily have made, rather, the HT-SMR was viewed as 'an additional tool' which could 'confirm a clinical impression', 'justify decisions' and give clinicians 'more confidence'.

### Adaptations and views about future use

Eight out of 12 psychiatrists thought that SMR should be used more widely. Suggestions to improve the system centred on making the system more 'user friendly' and 'intuitive' for psychiatrists, and integrated with existing computerised systems. Three psychiatrists also mentioned improving the accessibility to people with ID incorporating their views more formally in the medication review, for example, 'adding a weight to the [decision-support] algorithm based on patient preference'.

## DISCUSSION

### Main findings

There is a need to improve the quality of psychotropic medication use in people with ID, yet despite consensus guidelines of good practice[17 18] there has been relatively little work to investigate practical methods to achieve medication optimisation in this group. The current study introduced a structured medication review tool in community psychiatric services for adults with ID and demonstrates that it would be feasible to test outcomes in a definitive trial.

Perhaps owing to the scrutiny currently applied to psychotropic prescribing, clinical teams and psychiatrists that we approached were keen to take part in this research. Recruitment of people with ID to research can be challenging[19] but the number of participants we recruited was satisfactory, close to our original broad expectation, and the referral:recruitment ratio was high, indicating that the processes of participant identification, recruitment and consent were appropriate.

A key question was whether psychiatrists were able and willing to integrate use of the HT-SMR into their standard practice, given the demands on their time and numerous mandated clinical and administrative tasks. Uptake of the HT-SMR was good, though not universal; 3 psychiatrists did not use the tool and 11 people with ID who were recruited did not have a recorded medication review. The rate of missed appointments is higher in psychiatric clinics than in other medical specialties[20] and may be still higher in ID services, and missed appointments are likely to be one of the causes that limited the HT-SMR during the study period.

The HealthTracker-imputed MEI was tested as a potential future outcome measure. The MEI was correlated with the overall CGI-I and was lower (indicating a less favourable risk:benefit ratio) in those in whom medication changes were made compared with those in whom medication remained unchanged. The HealthTracker-imputed MEI showed sufficient variation between participants and had value as a practical support to psychiatrists in considering medication changes, though the survey data showed that this does not replace psychiatrists' clinical judgement. However, there may also be disadvantages to using a single measure of medication effect in those who receive polypharmacy as psychiatrists (and patients and their carers) may find it difficult to attribute changes to a specific medication.

This research involved relatively little commitment from participants with ID and the intervention appeared acceptable in view of the recruitment metrics and response to the evaluation questionnaire. Two-thirds of psychiatrists thought that the system should be used more extensively, indicating an overall favourable attitude. In order to maintain proximity to usual practice, we gave psychiatrists flexibility and few instructions on how to use the online system in their appointments, other than on how to enter data. There was clearly variation in how different psychiatrists approached the HT-SMR; positive feedback showed that some appreciated the systematic and comprehensive nature of the medication review and believed that it could facilitate a discussion with the person with ID. Negative comments referred to the perception that the structured review was inflexible and rigid. This may be related to natural variation in clinicians' consultation style and familiarity with incorporating standardised or structured elements to the consultation, though these are recommended in monitoring medication effects.[18 21]

Some psychiatrists reported disruption to the relational aspects of the consultation arising from the need to interact simultaneously with the computer screen and the person with ID and others who may attend the appointment. Electronic records are already used extensively in healthcare settings but use of technology as a more dynamic application may represent a more profound culture change and requires the development of new skills and ways of working. It is possible that digital interventions, if properly designed, can enhance communication between doctor and patient, for example, by incorporating augmentative and alternative communication methods.[22 23] Given that patient involvement and the opportunity for shared decision-making are fundamental to medication optimisation, the HT-SMR would benefit from incorporating a greater role for people with ID and their carers to amplify the patient voice. This could go some way to countering the lack of involvement that patients and their carers often describe when medication decisions are made.[24] Other opportunities to extend the remit of this system include patients or carers completing measures in advance of appointments in order to release consultation time for discussion and collaboration, particularly if the system was configured to prioritise the individuals' indication for medication and the most common adverse side-effects of the drug prescribed.

## Future work

A future clinical trial is needed to test if use of the HT-SMR contributes to medication optimisation. The HealthTracker-imputed MEI could be used as a primary outcome measure and should be supplemented by other measures of medication optimisation, including service utilisation, medication safety incidents and patient-reported outcomes, including decision self-efficacy and satisfaction. An economic evaluation is also necessary to determine the cost implications of the intervention; balanced against the additional resource and infrastructure necessary to deliver the HT-SMR are potential cost savings achieved through reductions in medication waste and in indirect costs related to adverse side-effects.

Wide-scale implementation of a system of structured medication review would create a powerful naturalistic data set of medication use, therapeutic impact and adverse side-effects that could be used both as a dashboard to monitor and benchmark prescribing practice, and for observational research in this group where there is a paucity of empirical data and little prospect of significant future controlled trials.

The STOMP campaign in England has so far not achieved discernible reductions in psychotropic prescribing to adults with ID.[25] Medication review, as an opportunity for critical reflection and discussion about medication, may act as a stimulus for change in prescribing that will ultimately improve medication outcomes. However, there are many influences on prescribing behaviour, including those acting on an individual level among patients, carers and clinicians,[26 27] as well as systemic factors that are likely to extend beyond the control of the prescriber, such as appropriately supported accommodation and social care

provision.[28] Thus, a medication review intervention can be only one element of a programme of medication optimisation and changing behaviour on a wider scale will require concerted action across health and social care sectors. One published report of a multicomponent intervention to reduce antipsychotic use has shown some success but was time-consuming, has not been replicated and lacks longer-term outcomes.[29] Future evidence-based complex interventions (of which structured medication review can be a part) that can work at scale should be underpinned by a theoretical framework that can identify the levers and barriers that are most likely to affect implementation.[30]

### Strengths and limitations of this study
This study was completed in real-world settings, included psychiatrists of different grades from several different services and a diverse group of participants, thereby increasing generalisability of the findings. We obtained estimates of important recruitment parameters and confirmed a successful recruitment strategy. Feedback has enabled us to identify aspects of the HT-SMR which require development to improve utility and enhance the potential for benefits of the intervention. The advantages of this mediation review were that it is relatively quick, self-explanatory and can be completed in a single patient contact, making it easier to integrate into the current models of care than other published medication review methods that are multistage and multiprofessional and more likely to encounter implementation barriers.[31 32] Being conducted by the psychiatrist, who is also the prescriber, the method avoids the pitfalls of non-prescriber-directed medication reviews in which as few as one-third of recommendations are actioned.[33]

This study also had limitations. We could not collect the characteristics of those who declined to participate in the research, and therefore do not know the total eligible population or whether certain groups were under-represented in our sample. Similarly, we do not know the number of appointments in which the system could have been used in but was not, and without this denominator we cannot report the rate of uptake. Attrition and clinician fatigue in using the online medication review may be an issue in a longitudinal study that was not addressed in this feasibility study, given the relatively short time period of the research. A single participant feedback question was chosen to minimise demands placed on participants but was inevitably limited in scope, and responses may have been subject to social desirability bias. Although logic suggests that the medication review would give patients and carers a greater opportunity for input in the process of medication decision-making, this was not formally tested and there was no method for gathering feedback from carers who may have been involved in the appointment and who play an important role in the medication process. We also included only a limited number of psychiatrists, and within this group some were more enthusiastic users of the HT-SMR than others. This introduces a further source of bias, as the results are largely driven by only a small number of psychiatrist users of the system.

### CONCLUSION
Medication review has the potential to improve individual medication outcomes as part of a wider programme of medication optimisation. The HT-SMR could be tested in a definitive trial after some refinement to improve integration with existing software and to fully embed patient and carer voice in the review process.

**Acknowledgements** The authors would like to thank staff in the clinical services who recruited participants and completed the HT-SMR, members of the Camden Synergy service user consultation group (Jill Huntesmith and Jackie McMorrow), and the research participants and their carers.

**Contributors** RS, AS, LM, NM, FF, PS and AH conceived and designed the study and contributed towards the conduct and management of the work. RS recruited participants. RS, FF and LM managed and analysed the data and AS, NM, PS and AH supervised this process. RS, AS, LM, NM, FF, PS and AH interpreted the results. RS wrote the first draft of the paper. AS, LM, NM, FF, PS and AH contributed to further drafts and read and approved the final manuscript.

**Funding** RS is funded by a National Institute for Health Research (NIHR) Doctoral Research Fellowship for this research project (Ref: DRF-2016-09-140).

**Disclaimer** This publication presents independent research funded by the National Institute for Health Research (NIHR). The views expressed are those of the authors and not necessarily those of the NHS, the NIHR or the Department of Health and Social Care

**Competing interests** FF reports that he is employed as Chief Technology Officer by HealthTracker Ltd—the company that has the copyright for the HealthTracker-based structured medication review (HT-SMR). The HT-SMR can be licensed to hospitals and healthcare settings. PS reports that he is a Director and shareholder of HealthTracker Ltd—the company that has the copyright for the HT-SMR. The HT-SMR can be licensed to hospitals and healthcare settings.

**Patient consent for publication** Not required.

**Ethics approval** The authors assert that all procedures contributing to this work comply with the ethical standards of the relevant national and institutional committees on human experimentation and with the Helsinki Declaration of 1975, as revised in 2008. All procedures involving patients were approved by the London Bridge Research Ethics Committee (ref: 18/LO/1112).

**Provenance and peer review** Not commissioned; externally peer reviewed.

**Data availability statement** No data are available. All data relevant to the study are included in the article or uploaded as supplementary information.

**ORCID iDs**
Rory Sheehan http://orcid.org/0000-0002-4164-9661
Nicola Morant http://orcid.org/0000-0003-4022-8133
Paramala Santosh http://orcid.org/0000-0003-4830-5893

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
