## [Reviewer comments · BMJ Open]

ARTICLE DETAILS

TITLE (PROVISIONAL)	A structured medication review tool to promote psychotropic medication optimisation for adults with intellectual disability: feasibility study
AUTHORS	Sheehan, Rory; Strydom, André; Marston, Louise; Morant, Nicola; Fiori, Federico; Santosh, Paramala; HASSIOTIS, ANGELA

VERSION 1 – REVIEW

REVIEWER	Maria Valdovinos Drake University, USA
REVIEW RETURNED	11-Sep-2019

GENERAL COMMENTS	I appreciate the opportunity to review the manuscript titled, "A structured medication review tool to promote psychotropic medication optimisation for adults with intellectual disability: feasibility study." Indeed, the prevalence of psychotropic use, the prevalence of polypharmacy, and the doses prescribed makes the tool described in the paper quite timely and necessary. That being said, I did find that I had questions while reviewing the paper that influenced my decision to recommend major revisions before being accepted for publication. I've outlined my questions below. 1. As I was reading the paper, I was struck by the question, "who are the participants?" The authors identify the individuals with intellectual disability as the participants but it also seems to me that the psychiatrists are also participants. It is their behavior that is being evaluated (i.e., prescribing) and the intervention is designed to impact that behavior, not to necessarily change the behavior of the individuals prescribed the medication. I wonder if the psychiatrists gave consent for their participation and what that consent process was like.2. The methods state that the psychiatrist completed the CGI for each medication as a further measure of medication effect. How was conjunctive or polypharmacy handled? How is each medication assessed when medications may be prescribed in combination? It is unclear if there were patients with this medication regimen and how these situations were addressed.3. It is unclear what symptoms were being treated and it seems that perhaps these data were not collected by the Health Tracker? Though commonly used, the CGI is not a specific tool and relies heavily on the subjective interpretation of the psychiatrist as often informed by caregivers. Given caseloads that psychiatrists often have, I wonder if there were more observable symptoms psychiatrists were allowed to note in the tool that they could reflect upon during subsequent visits.4. It is stated that researchers trained psychiatrists on the system but there is no information on what that training entailed (e.g.,
--

	verbal instruction, role play, simulated patient, practice entries, etc.). 5. In the limitations, I think some consideration regarding the potential bias in the results is justified. Given the few number of psychiatrist who participated and that it appears clinics 1, 2, and 5 are somewhat over-represented in the results, might it be fair to say the results could have been influenced by the use of certain psychiatrists rather than the instrument itself? To that end, knowing how many patients per psychiatrists and the distribution would be helpful in determining the tools impact and potential reflection of bias in the results. 6. In the results, it is stated that those who had a medication change following the review had MEI's that were significantly lower. Were there particular medication changes that impacted MEI's (e.g., increases in dose, decreases, addition of new medication, withdrawal, etc.) 7. Related to my earlier comment on training on the tool, in the discussion the authors state they gave few instructions on use of the tool. In the absence of instructions, it seems a tool is just a tool. I was left wondering if the independent variable is the tool itself or how one is taught to use the tool to guide practice? Entering data, it would seem, is one element of using the tool but the other is interpreting the data? 8. I do agree that having caregivers complete measures in advance could be impactful, particularly if the tool is individualized and designed to auto-populate with common adverse side effects associated with a given individual's medication regimen and the changes in behavior or symptoms that are being monitored to determine medication effectiveness.
--	--

REVIEWER	Matthias Schützwahl Department of Psychiatry and Psychotherapy TU Dresden Germany
REVIEW RETURNED	11-Sep-2019

GENERAL COMMENTS	Many thanks for the opportunity to review the manuscript with the ID bmjopen-2019-033827. The intervention under study aims to optimise the psychopharmacological treatment of people with intellectual disabilities. This is an important topic that will be of interest to many readers given the observed over-medication of this group of people. The manuscript is clearly structured and well-written. In my opinion, only minor revisions are necessary:  - Throughout the manuscript, frequencies should be formulated as concretely as possible (cf. p. 18, line 11, "a small number"). - Overall, the authors do not seem to make a stringent decision as to who the study participants are. In the abstract, but not later on, the participating psychiatrists are correctly referred to as study participants. - Information on the 15 participating psychiatrists is missing. Information e.g. on age and work experience would be helpful. - In the limitations it should be reported that it is not possible to investigate the influence of these data due to the small number of participating psychiatrists.
---

VERSION 1 – AUTHOR RESPONSE

Reviewer(s)' Comments to Author:

Reviewer: 1

Reviewer Name: Maria Valdovinos

Institution and Country: Drake University, USA Please state any competing interests or state 'None declared': None declared

Please leave your comments for the authors below

I appreciate the opportunity to review the manuscript titled, "A structured medication review tool to promote psychotropic medication optimisation for adults with intellectual disability: feasibility study." Indeed, the prevalence of psychotropic use, the prevalence of polypharmacy, and the doses prescribed makes the tool described in the paper quite timely and necessary. That being said, I did find that I had questions while reviewing the paper that influenced my decision to recommend major revisions before being accepted for publication. I've outlined my questions below.

1. I was reading the paper, I was struck by the question, "who are the participants?" The authors identify the individuals with intellectual disability as the participants but it also seems to me that the psychiatrists are also participants. It is their behavior that is being evaluated (i.e., prescribing) and the intervention is designed to impact that behavior, not to necessarily change the behavior of the individuals prescribed the medication. I wonder if the psychiatrists gave consent for their participation and what that consent process was like.

At each research site, a senior clinician was invited to take part in the study and acted as a local investigator. Following formal approval from the organisation within which they worked, other members of the medical team at each site were given information about the project and invited to take part. Those who agreed were provided with login details for the HealthTracker website and were trained (described below). They were then able (but not compelled) to use the HT-SMR during appointments in the study period in people with ID who had been recruited to the study.

At the end of the study period the clinicians were invited to take part in an anonymous feedback exercise (an online survey). Preliminary material gave information regarding the purpose of the survey and the potential for use of quotations in the study report. This survey did not include the collection of personal data.

The research team were available to be contacted at all points, should the clinicians have had any questions or concerns. Study processes were reviewed and approved by the NHS ethics committee and the UK Health Research Authority.

2. The methods state that the psychiatrist completed the CGI for each medication as a further measure of medication effect. How was conjunctive or polypharmacy handled? How is each medication assessed when medications may be prescribed in combination? It is unclear if there were patients with this medication regimen and how these situations were addressed.

All measures (GCI-I, POTR) were conducted for each medication that was reviewed. Where people received polypharmacy, the clinician may have chosen to focus the review on a single medication at the appointment, or to review more than one medication at the same appointment, in this case completing separate scales for each medication. We realise that it might be challenging to ascribe change to the effect of one medication where a person takes several drugs simultaneously - this is a function of the complexity of the patient group and the high rates of polypharmacy in real-life practice and is an issue commonly faced by clinicians. The HT-SMR is a tool that allows clinicians to clearly and consistently document their impression of a medication effect but in itself is not an answer for clinical and therapeutic complexity.

We have added to the manuscript in response to this point (method – page 9) and also now included the limitations of this approach (discussion – page 20/21).

3. It is unclear what symptoms were being treated and it seems that perhaps these data were not collected by the Health Tracker? Though commonly used, the CGI is not a specific tool and relies heavily on the subjective interpretation of the psychiatrist as often informed by caregivers. Given caseloads that psychiatrists often have, I wonder if there were more observable symptoms psychiatrists were allowed to note in the tool that they could reflect upon during subsequent visits.

As part of the data entry for the HealthTracker system, clinicians were able to add the patient's primary psychiatric diagnosis from a dictionary based on the International Classification of Diseases. These data are presented in table 1 (results – page 13).

The profile of treatment response (POTR) is a comprehensive list of target symptoms and possible adverse medication effects that are grouped by type – this is where the presence of symptoms or side-effects is recorded by the clinician. It was not possible for clinicians to make their own, free-text notes but we agree that this may be a useful addition to future iterations of the system.

4. It is stated that researchers trained psychiatrists on the system but there is no information on what that training entailed (e.g., verbal instruction, role play, simulated patient, practice entries, etc.).

Instructions on use of the HT-SMR were focused on the practical aspects of using the system. This included finding and logging onto the host website, creating a patient profile, assigning the relevant scales, inputting data to these scales, and viewing the Modified Efficacy Index matrix.

This training was delivered to all clinicians by the lead author in small groups, and included a demonstration of use of the HT-SMR using a fictional example. The training session was arranged at a time and place most convenient for the clinicians, and as such, the practicalities varied slightly.

We have added to the manuscript to provide more detail of the training (method – page 9).

5. In the limitations, I think some consideration regarding the potential bias in the results is justified. Given the few number of psychiatrist who participated and that it appears clinics 1, 2, and 5 are somewhat over-represented in the results, might it be fair to say the results could have been influenced by the use of certain psychiatrists rather than the instrument itself? To that end, knowing how many patients per psychiatrists and the distribution would be helpful in determining the tools impact and potential reflection of bias in the results.

We agree with the limitations that the reviewer highlights, and which are inherent in small-scale feasibility studies. We have added the small numbers involved to the limitations section of the manuscript (discussion – page 25).

We have now added how many medication reviews per psychiatrist and the distribution (page 13).

6. In the results, it is stated that those who had a medication change following the review had MEI's that were significantly lower. Were there particular medication changes that impacted MEI's (e.g., increases in dose, decreases, addition of new medication, withdrawal, etc.)

It is not possible for us to determine the influence of the HT-SMR on medication effect, as most reviews were one-off events and therefore give only cross-sectional data. These data show that the modified efficacy index was correlated with the clinician decision to change medication at the time of the medication change. Lower efficacy index scores (which indicate a less favourable benefit:side-effect ratio) were more likely to be associated with change in medication, but the impact of this change could only be answered in a later efficacy study with longer follow-up times.

7. Related to my earlier comment on training on the tool, in the discussion the authors state they gave few instructions on use of the tool. In the absence of instructions, it seems a tool is just a tool. I was left wondering if the independent variable is the tool itself or how one is taught to use the tool to guide practice? Entering data, it would seem, is one element of using the tool but the other is interpreting the data?

Instructions on use of the tool were limited to the use of the tool itself, rather than direction in how to incorporate the tool into appointments or how to use the MEI output.

The proposed logic behind the HT-SMR is that completing the tool gives a clear place in an appointment for a specific discussion about medication, and that this discussion is comprehensive and rigorous. The MEI score could lead to a more objective evaluation of medication effects and the presentation of this in colour-coded graphical could stimulate a discussion between doctor and patient/carer about medication decisions. We were not at liberty to produce formal guidelines on interpreting the MEI and decision-making responsibility in consultations did not change.

8. I do agree that having caregivers complete measures in advance could be impactful, particularly if the tool is individualized and designed to auto-populate with common adverse side effects associated with a given individual's medication regimen and the changes in behavior or symptoms that are being monitored to determine medication effectiveness.

Thank you, we have added to the manuscript with this point (discussion – page 22).

Reviewer: 2

Reviewer Name: Matthias Schützwahl

Institution and Country:

Department of Psychiatry and Psychotherapy TU Dresden Germany Please state any competing interests or state 'None declared': None declared

Please leave your comments for the authors below

Many thanks for the opportunity to review the manuscript with the ID bmjopen-2019-033827.

The intervention under study aims to optimise the psychopharmacological treatment of people with intellectual disabilities. This is an important topic that will be of interest to many readers given the observed over-medication of this group of people.

The manuscript is clearly structured and well-written.

In my opinion, only minor revisions are necessary:

- Throughout the manuscript, frequencies should be formulated as concretely as possible (cf. p. 18, line 11, "a small number").

We have now given specific figures throughout the reporting of the clinician feedback (pages 16-19).

- Overall, the authors do not seem to make a stringent decision as to who the study participants are. In the abstract, but not later on, the participating psychiatrists are correctly referred to as study participants.

We have amended the paper accordingly and made distinctions more clearly where we are referring to psychiatrists who took part in the study and people with ID who took part.

- Information on the 15 participating psychiatrists is missing. Information e.g. on age and work experience would be helpful.

We have now included some general information on the psychiatrists who took part in the study (results – page 12).

- In the limitations it should be reported that it is not possible to investigate the influence of these data due to the small number of participating psychiatrists.

We have now addressed this point which was also raised by reviewer 1 (discussion – pages 24-25).

VERSION 2 – REVIEW

REVIEWER	Matthias Schützwahl Department of Psychiatry and Psychotherapy, TU Dresden, Germany
REVIEW RETURNED	16-Oct-2019
GENERAL COMMENTS	The authors have sufficiently addressed all comments so that the manuscript is now recommended for publication.